# Peripheral nerve resident macrophages share tissue-specific programming and features of activated microglia

Peter L. Wang[1,2,3], Aldrin K. Y. Yim[2,3], Ki-Wook Kim[1], Denis Avey[2], Rafael S. Czepielewski [1], Marco Colonna[1], Jeffrey Milbrandt[2✉] & Gwendalyn J. Randolph [1✉]

Whereas microglia are recognized as fundamental players in central nervous system (CNS) development and function, much less is known about macrophages of the peripheral nervous system (PNS). Here, by comparing gene expression across neural and conventional tissue-resident macrophages, we identified transcripts that were shared among neural resident macrophages as well as selectively enriched in PNS macrophages. Remarkably, PNS macrophages constitutively expressed genes previously identified to be upregulated by activated microglia during aging, neurodegeneration, or loss of Sall1. Several microglial activation-associated and PNS macrophage-enriched genes were also expressed in spinal cord microglia at steady state. We further show that PNS macrophages rely on IL-34 for maintenance and arise from both embryonic and hematopoietic precursors, while their expression of activation-associated genes did not differ by ontogeny. Collectively, these data uncover shared and unique features between neural resident macrophages and emphasize the role of nerve environment for shaping PNS macrophage identity.

[1] Division of Immunobiology, Department of Pathology and Immunology, Washington University School of Medicine, St. Louis, MO 63110, USA. [2] Department of Genetics, Washington University School of Medicine, St. Louis, MO 63110, USA. [3]These authors contributed equally: Peter L. Wang, Aldrin K. Y. Yim. ✉email: jmilbrandt@wustl.edu; gjrandolph@wustl.edu

The significant role of resident neural macrophages in neuroinflammation and disease progression is increasingly appreciated in mouse models and individuals with neurodegeneration[1–3]. Such advances, which largely rely on the interpretation of data from transcriptional analyses and human genome-wide association studies of Alzheimer's disease (AD) and other neurodegenerative conditions, have led to critical findings about cellular and molecular processes underlying such diseases[4–6]. Most of these studies, however, have focused on resident macrophages in the brain (microglia) and, to a lesser extent, the spinal cord. Meanwhile, the transcriptional identity and functions of resident macrophages in the peripheral nervous system (PNS) remain mostly unknown.

The PNS consists of a multitude of neuronal networks that relay motor and sensory information between the central nervous system (CNS) and the rest of the body[7]. Although it has the capacity to regenerate, the PNS is also prone to injury and degeneration[8]. Studies of PNS injury have shown that PNS macrophages play important roles for debris clearance, pain development, and regeneration[9–11]. While the contribution of recruited monocytes cannot be excluded, these studies demonstrate the importance of PNS macrophages in nerve injury. Understanding the roles of these cells in homeostasis and disease may be broadly beneficial for resolving neuroinflammation.

In addition to monocyte-derived macrophages in nerve injury, there are also resident macrophages in the PNS at steady state[12,13]. While their residence in neuronal tissues is inherently microglia-like, PNS macrophages exist within a unique peripheral nerve microenvironment. Moreover, although it is known that CNS microglia are derived from the yolk sac during embryogenesis[14], the ontogeny and developmental requirements of PNS macrophages remains unclear. Considering the growing interest in how tissue environment and ontogeny contribute to microglial identity and function in neurological diseases, we investigated these questions in PNS macrophages.

Here we show that self-maintaining PNS-resident macrophages are a distinct population with transcriptional signatures relating to specific functions in neuronal tissue. PNS macrophages significantly express genes that resemble not only homeostatic microglia but also activated microglia from aging and neurodegenerative conditions. We found that PNS macrophages originate from both embryonic and hematopoietic sources, and exhibit partial dependence on interleukin (IL)-34, an alternative ligand for colony-stimulating factor 1 receptor (CSF1R) that is important for the development and maintenance of microglia[15,16]. With the exception of Ccr2 expression, transcriptional signatures in PNS macrophage were largely similar between embryonic and hematopoietic sources, suggesting that nerve environment controls their identity at steady state.

## Results

### Resident macrophages of the PNS
To examine resident macrophages in the PNS, we imaged a variety of nerve types at steady state using CX3CR1$^{GFP/+}$ reporter mice. In these mice, green fluorescent protein (GFP) effectively labels microglia and has been shown to label nerve-associated macrophages in adipose, skin, lung, and enteric tissues[17–21]. CX3CR1$^{GFP/+}$ cells were found in dorsal root ganglia (DRG), vagal nerves (VNs), cutaneous intercostal fascial nerves (FNs), and sciatic nerves (SNs) (Fig. 1a). CX3CR1$^{GFP/+}$ cells were located in the endoneurium (Fig. 1b) and expressed CSF1R, also known as CD115 (Fig. 1c). Using flow cytometry, we found that CX3CR1$^{GFP/+}$ cells also expressed the common macrophage marker CD64 (FcγR1)[22] and intermediate levels of CD45 (Fig. 1d–f). Thus, CX3CR1$^{GFP/+}$ cells in peripheral nerves are indeed macrophages with some

resemblance to CNS microglia based on both endoneurial localization and surface marker expression.

To determine whether PNS macrophages depend on circulating precursors or are maintained via local signals, we performed parabiosis in CD45.1+ wild type and CD45.2+ Lyz2Cre × tdTomato$^{fl/fl}$ mice, and assessed the extent to which cells circulating from the parabiotic partner gave rise to PNS macrophages. Ten weeks after joining the parabionts, we found minimal exchange of PNS macrophages in all of the nerve types examined, whereas blood T cells and monocytes exchanged robustly (Fig. 1g and Supplementary Fig. 1). Indeed, most of the tdTomato+ cells that could be seen in the wild-type parabiont were localized to the tissue surrounding the nerves (Fig. 1h). We also performed pulse-chase labeling of PNS macrophages using tamoxifen-inducible CSF1R$^{Mer-iCre-Mer}$ × dTomato$^{fl/fl}$ mice. In these mice, tdTomato expression persists in self-maintaining cells, but not in monocytes, which mostly turn over by 3–4 weeks after tamoxifen removal[23,24]. Heterozygous mice were fed tamoxifen diet for 4 weeks and then switched to normal diet (Fig. 1i). Just following tamoxifen removal, 96% of PNS macrophages (pooled from all PNS sites), 99% of CNS microglia, and 100% of blood monocytes were tdTomato+ (Fig. 1j–l and Supplementary Fig. 2). Whereas only 20% of nonclassical and classical monocytes were still tdTomato+ by 3 weeks after tamoxifen removal, 98% of CNS microglia and 95% of pooled PNS macrophages remained labeled up to 8 weeks following tamoxifen removal (Fig. 1l and Supplementary Fig. 2). Taken together, these results indicate that PNS macrophages are mostly self-maintained in adult mice.

### Transcriptional characterization of PNS macrophages
As we and others have previously demonstrated, unique gene expression profiles can be obtained in tissue-resident macrophage populations across tissue types[22,25]. To identify signature genes in peripheral nerve macrophages, we performed bulk RNA sequencing (RNA-seq) to compare purified PNS resident macrophages sorted from DRG, VN, cutaneous intercostal FN, and SNs (Supplementary Fig. 3) with CNS microglia from the brain and spinal cord, as well as previously characterized conventional macrophage populations from the spleen, peritoneal cavity, and lungs. Global transcriptomic analysis revealed similarities within resident neural macrophages from both PNS and CNS, with PNS macrophages clustering more closely to CNS microglia than to conventional macrophages (Fig. 2a and Supplementary Data 1). A substantial number of genes were uniquely enriched in PNS macrophages and CNS microglia compared with the other tissue-resident macrophages, including microglial signature genes Tmem119, P2ry12, Siglech, Trem2, and Olfml3 (Fig. 2b, c). PNS macrophage-specific genes were also identified (Fig. 2b).

To determine potential functions of PNS macrophages associated with their shared and unique gene expression profiles, we performed Gene Ontology analysis on transcripts that were common between PNS macrophages and CNS microglia, and those that were specific to PNS macrophages (Supplementary Data 2). Consistent with the idea that PNS macrophages may share functions with CNS microglia, pathway analysis identified functions including synaptic plasticity, microglial motility, and positive regulation of neurogenesis (Fig. 2d). Pathways that were unique to PNS macrophages included angiogenesis, collagen fibril organization, regulation of BMP signaling, and peripheral nerve structural organization and axon guidance (Fig. 2e).

Next, we identified transcripts that were 4-fold or more enriched in CNS microglia and PNS macrophages relative to their expression in all other conventional macrophage populations (Fig. 2f). These upregulated genes included Abhd6, Ophn1, P2rx7,

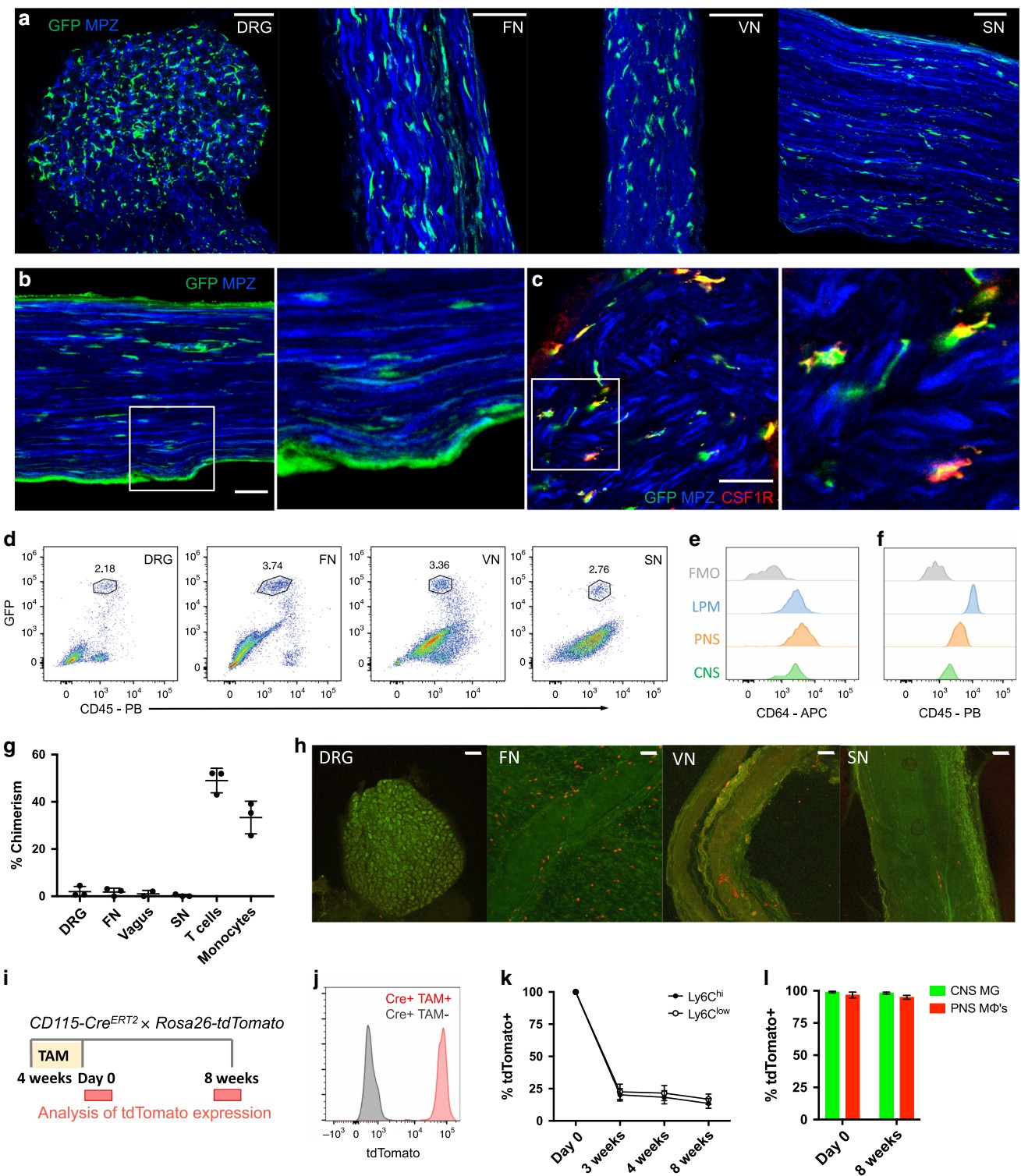

*Pld1, Sgce, Tgfbr1, Tfpi,* and *Tmem173.* We also identified several genes that were downregulated in CNS microglia and PNS macrophages (Supplementary Fig. 4). Notably, the transcriptional regulator that defines microglial identity and function, *Sall1,* was not expressed by PNS macrophages (Fig. 2c). Modest detection of *Sall1* in the DRG could not be corroborated by further analysis (Supplementary Fig. 5). This may reflect unique adaptations in PNS and CNS macrophages. Indeed, we identified 72 genes, including *Sall1,* which were highly specific to CNS microglia (Supplementary Fig. 6).

To examine unique gene expression in PNS macrophages, we identified transcripts that were at least fourfold higher or lower in PNS macrophages compared with other resident macrophages, including CNS microglia (Fig. 2g and Supplementary Fig. 7). We found 48 genes specifically enriched in PNS macrophages, including *Aplnr, Cp, Il1rl1, Maoa, Pla2g2d,* and *St8sia4,* as well as interferon-induced genes *Ifi202b, Ifi211,* and *Oas2.* We also identified *Ms4a14, Ms4a4a, Ms4a4c, Ms4a6c,* and *Ms4a7.* These signatures reveal unique transcriptional programming in PNS macrophages and may

**Fig. 1 Identification and characterization of PNS resident macrophages. a–c** Representative confocal imaging of peripheral nerves from CX3CR1[GFP/+] MPZ[tdTomato] mice (tomato from MPZ depicted here in blue). **a** Images of whole-mount dorsal root ganglia (DRG), cutaneous fascial (FN), vagal (VN), and sciatic nerves (SN) isolated from CX3CR1[GFP/+] MPZ[tdTomato] mice. Scale bars are 50 μm. **b** Endoneurial localization of GFP+ cells in longitudinal sections of sciatic nerves from CX3CR1[GFP/+] MPZ[tdTomato] mice. **c** CSF1R (red) and CX3CR1[GFP] (green) colocalization in sciatic nerve cross sections. Scale bars are 50 μm. **d** Flow cytometric gating of CX3CR1[GFP/+] cells from peripheral nerve tissues. **e, f** Representative expression of CD64 and CD45 in CX3CR1[GFP/+] cells compared with brain microglia, large peritoneal macrophages (LPMs), and fluorescence minus one (FMO) control. **g** Flow cytometric quantification of CD45.1 and CD45.2 chimerism in blood (total T cells or total monocytes) and nerves from three pairs of wild-type (CD45.1) and Lyz2Cre tdTomato (CD45.2) parabionts 10 weeks after joining. **h** Representative imaging in peripheral nerves from wild-type parabiont; Scale bars are 100 μm. **i–l** Analysis of tdTomato expression in tamoxifen-pulsed CSF1R[Mer-iCre-Mer] × Rosa26-tdTomato mice. **i** Tamoxifen delivery schematic for fate mapping. Mice were given tamoxifen diet for 4 weeks and analyses for PNS macrophages and CNS microglia were performed at 0 days and 8 weeks after tamoxifen diet removal. **j** tdTomato expression by genotype from combined peripheral nerves in tamoxifen-fed mice. **k** Flow cytometric quantification of Ly6c high and Ly6C low monocytes from mice bled at 0 days, 3 weeks, 4 weeks, and 8 weeks after tamoxifen removal. **l** Flow cytometric quantification of CNS microglia (brain and spinal cord) and PNS macrophages (pooled from DRG, fascial nerve, vagus nerve, and sciatic nerve to increase yield in analysis) 0 days and 8 weeks following tamoxifen removal. Data are mean ± SEM (n = 3 mice per time point). Source data are available as a Source Data file.

provide clues about their involvement in neuronal health and disease.

We next investigated transcriptional differences within PNS macrophage populations (Supplementary Fig. 8). Using a fourfold cutoff, we identified 24 genes enriched in SN macrophages, 23 genes enriched in cutaneous intercostal (fascial) nerve macrophages, and 12 genes enriched in VN macrophages. We observed similar numbers of downregulated genes in each population. We also compared nerve-resident macrophages to those residing within the dorsal root ganglion and found that they were significantly different, with many differentially expressed transcripts. Therefore, we re-analyzed this data using a more stringent 8-fold cutoff and identified 79 upregulated genes and 52 downregulated genes. These results suggest that although PNS macrophages are transcriptionally similar, significant differences exist between those adjacent to axons and those residing close to neuronal cell bodies.

**PNS macrophages express microglial activation genes**. To investigate differentially expressed genes (DEGs) within resident neural macrophages, we refined our analysis to CNS microglia and PNS macrophages (Fig. 3a). We identified 396 genes enriched in PNS macrophages and 180 genes enriched in CNS microglia (Supplementary Data 3). As the upregulation of *MS4A* family and interferon-induced genes has been reported to characterize aged and neurodegenerative disease-associated microglia[6,26,27], we wondered whether PNS macrophages expressed other genes associated with microglial activation. By cross-referencing published data[6], we determined the number of connections between disease-associated genes that were upregulated in activated microglia from aging, phagocytic, and neurodegenerative conditions and neural macrophage-enriched genes from either PNS macrophages or CNS microglia (Fig. 3b). We found 148 disease-associated genes that were enriched in PNS macrophages compared to 17 that were enriched in CNS microglia (Fig. 3b and Supplementary Data 4). From the highest connectivity groups 6–4, we identified 25 genes that were significantly higher in PNS macrophages, including *Ch25h*, *Anxa2*, *Cd52*, *Ifitm3*, *Cybb*, *Fxyd5*, *Igf1*, and *Apoe* (Fig. 3c).

Microglia lacking certain genes for homeostatic regulation have also been found to shift their gene expression towards an activated phenotype[28,29]. *Sall1* has been identified as a transcriptional regulator of microglia identity and function, with *Sall1*[−/−] microglia resembling inflammatory phagocytes[28]. As PNS macrophages did not express *Sall1* at steady state, we examined whether genes that are reportedly dysregulated in *Sall1*[−/−] microglia showed the same pattern of expression in PNS macrophages. Indeed, we found a high correlation between genes enriched in PNS macrophages and *Sall1*[−/−] microglia, including *Apoe*, *H2-Aa*, *Ms4a7*, *Clec12a*, *Aoah*, and *Cybb* (Fig. 3c). These

data suggest that, beyond the difference in *Sall1*-driven gene expression, PNS macrophages may share common genetic regulators with CNS microglia.

As microglia activation may occur under cell sorting conditions[30], we were concerned that the activation signature in PNS macrophages might be attributed to tissue preparation. Thus, we stained freshly fixed peripheral nerves for Clec7a and MHCII, which are induced in microglia across many activation states[6,29,31]. Resting PNS macrophages were clearly marked by Clec7a and MHCII (Fig. 3e), suggesting that the signature obtained in PNS macrophages is not a technical artifact that arose from activation induced by disaggregation of tissue. Taken together, these data show that PNS macrophages constitutively express a wide array of microglial activation genes, including genes upregulated by microglia after loss of *Sall1*, implying the possibility of a shared microenvironment-sensitive regulation of gene expression in resident neural macrophages.

**PNS to CNS zonation in resident neural macrophages**. Given the difference in gene expression between PNS macrophages and CNS microglia, we wanted to further discriminate the influence of microenvironment on neural macrophage identity. Thus, we examined spinal cord microglia, which reside in a distinct microenvironment from brain microglia. Interestingly, we found a set of genes that were high in PNS macrophages, intermediate to high in spinal cord microglia and low in brain microglia (PNS to CNS zonation) (Fig. 4a). These included PNS macrophage-specific genes (*Cp*, *Il1rl1*, *Maoa*, and *Cdr2*), microglial activation genes (*Clec7a*, *Spp1*, *Lpl*, *Axl*, *Ms4a4c*, and *Ms4a6c*), interferon-induced genes (*Ifi204*, *Ifi207*, *Ifi209*, and *Oasl2*), mitochondrially encoded genes (*mt-Nd1*, *mt-Nd2*, *mt-Nd4*, *mt-Nd5*, and *mt-Nd6*), and several transcription factors (*Hivep2*, *Zfp704*, and *Rbpj*) (Fig. 4b, c). We confirmed Clec7a expression in spinal cord microglia by immunostaining (Fig. 4c). Importantly, microglia from spinal cord and brain did not significantly differ by expression of homeostatic genes *Sall1*, *Olfml3*, and *Tmem119* (Supplementary Fig. 9). In fact, certain microglial genes, including *Tgfbr1* and *P2ry12*, were more highly expressed in spinal cord compared with brain microglia. These findings suggest that transcriptional programs underlying PNS macrophages and activated microglia may be present during normal physiological conditions and further support the role of nerve environment for specifying neural macrophage identity.

**Ontogeny of PNS resident macrophages**. In addition to microenvironmental cues, ontogeny may also play an important role for specifying microglial identity and function. Specifically, it has been shown that, compared to naturally occurring microglia with yolk sac origin, monocyte- and hematopoietic stem cell (HSC)-

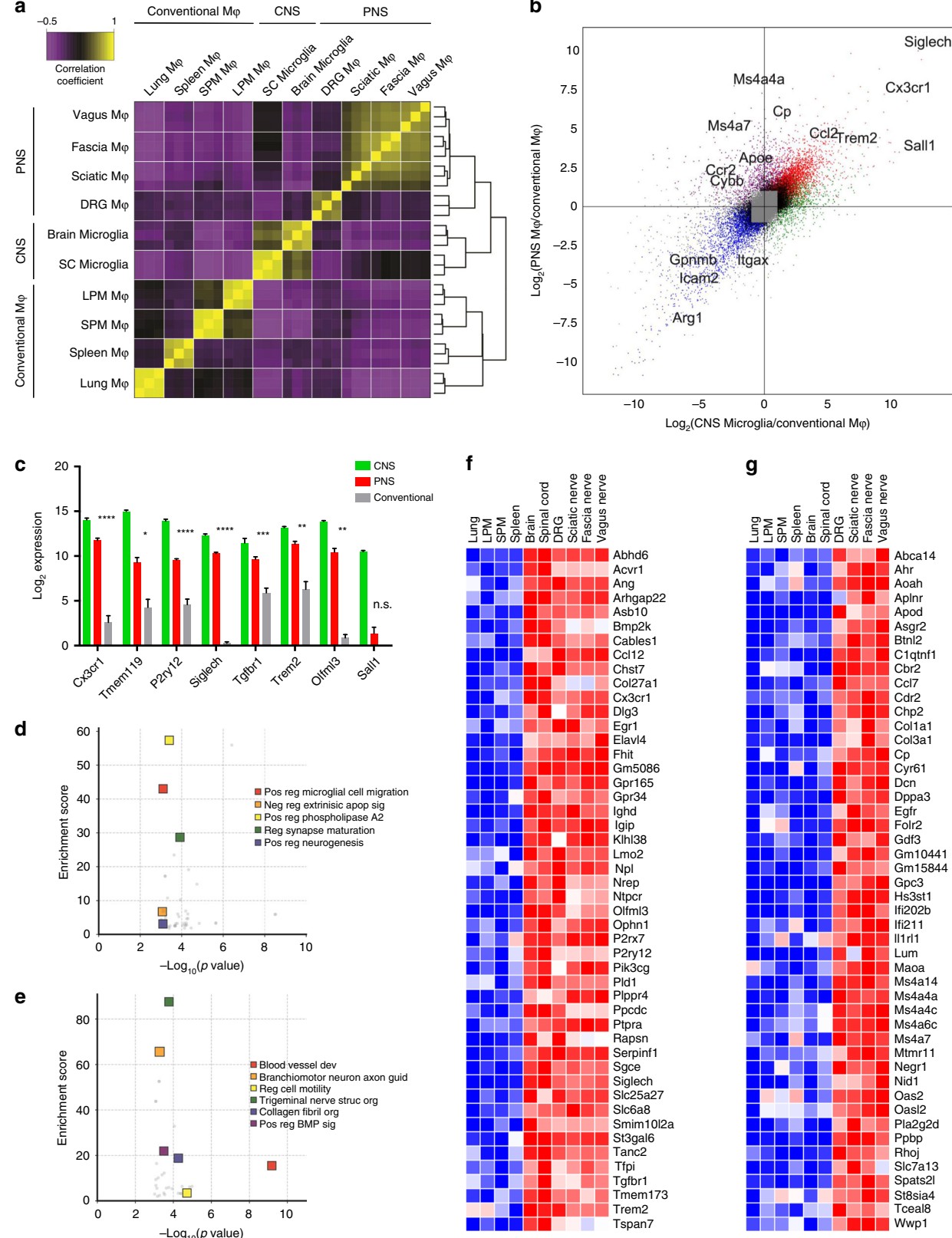

derived microglia display a more activated signature[32,33]. Thus, we sought to determine PNS macrophage ontogeny by examining Flt3Cre × LSL-YFP[fl/fl] reporter mice (Fig. 5a), a strain that labels fetal monocyte and adult HSC-derived hematopoietic multipotent progenitors and their progeny[34,35]. We found that ~26% of PNS macrophages across nerve types appeared to be embryonically

derived (YFP−) and 74% of PNS macrophages were HSC-derived (YFP+) (Fig. 5b and Supplementary Fig. 10).

To further examine the contribution of embryonic precursors to PNS macrophages, we performed fate mapping using CSF1R[Mer-iCre-Mer] × tdTomato[fl/fl] × CX3CR1[GFP/+] double repor-ter mice, which allows simultaneous visualization of tamoxifen-

**Fig. 2 PNS macrophages express microglial transcripts as well as a unique signature. a** Sample correlation plot showing global transcriptomic analysis and hierarchical clustering of resident macrophages from PNS, CNS, and conventional macrophages. Each box represents one replicate. Three replicates comprising up to 20 mice per replicate were included for each population. **b** Visualization of PNS macrophage unique transcripts (upper quadrant), CNS microglia unique transcripts (right quadrant), shared transcripts between PNS macrophages and CNS microglia (diagonal, top right quadrant), and conventional macrophages (bottom right quadrant). **c** Expression of microglial core transcripts in combined PNS macrophages compared with combined conventional macrophages. Multiple *t*-tests. Data are mean ± SEM. *$P < 0.05$, **$P < 0.01$, ***$P < 0.001$, ****$P < 0.0001$. **d** GO analysis of genes enriched in PNS macrophages and CNS microglia. **e** GO analysis of genes enriched in PNS macrophages. **f** Transcripts expressed at least fourfold higher in PNS macrophages and CNS microglia than conventional tissue-resident macrophages ($p \leq 0.05$). **g** Transcripts expressed at least fourfold higher in PNS macrophages than CNS microglia or conventional macrophages ($p \leq 0.05$). **f, g** Each box represents average of three replicates. Abbreviations: LPM, large peritoneal macrophage; SPM, small peritoneal macrophage. Source data are available as a Source Data file.

pulsed CSF1R-expressing macrophages and resident Cx3cr1+ macrophages. After giving tamoxifen at embryonic day 8.5 (E8.5), a developmental time point that labels yolk-sacderived macrophages, we checked newborn pups for labeling. The PNS was fully populated with CX3CR1$^{GFP/+}$ macrophages at birth. In line with previous observations[36], we observed partial labeling in 42% of brain microglia (Fig. 5c, d). Labeling in PNS macrophages was 8% from SN and DRG, or roughly one-fifth of the entire PNS macrophage population when normalized to microglia (Fig. 5c, d). Less than 1% of macrophages were tdTomato+ in spleen, kidney, and blood, and only 3% of lung macrophages were tdTomato+ (Fig. 5c, d and Supplementary Fig. 11). These results demonstrate that the PNS is fully populated by macrophages at the time of birth and that a subset of these cells are derived from yolk sac progenitors.

Although monocyte entry into injured nerves depends on Ccr2[37,38], it is not known whether Ccr2 is required for seeding or maintaining the PNS macrophage niche. To address this question, we quantified CSF1R+ macrophages in SNs of Ccr2 knock-in (Ccr2$^{GFP/GFP}$) and control (Ccr2$^{GFP/+}$) mice. We observed no difference in PNS macrophage numbers between Ccr2 knockouts and controls (Fig. 5e–g). However, although GFP+ cells were present in SNs from Ccr2$^{GFP/+}$ mice, GFP+ cells were almost entirely absent from Ccr2$^{GFP/GFP}$ mice (Fig. 5e, f and Supplementary Fig. 12). These observations suggest that Ccr2-dependent monocyte entry is not required to fill or maintain the PNS macrophage niche, but may contribute to a modest subset of PNS macrophages at steady state.

Given the resemblance between microglia and PNS macrophages, we wondered if IL-34, an alternative ligand for CSF1R that contributes to CNS microglia homeostasis[15,16], likewise governs PNS macrophage numbers. We quantified PNS macrophages of IL-34 deficient mice (IL-34$^{LacZ/LacZ}$) by flow cytometry and imaging. In the SN, the fraction of CD45$^+$ cells in the disaggregated nerve was reduced by a 30% (Fig. 5h–i), apparently owing to more than 50% reduction in PNS macrophage density per nerve area, as examined by imaging the intact nerve (Fig. 5j–k). In the DRG, we observed a reduction in macrophage density of more than 35% (Supplementary Fig. 13), underscoring significant dependence on IL-34 in multiple PNS sites. Indeed, this fraction of loss resembles that observed in the CNS microglial population[15,16].

In the SN, we checked expression of IL-34 mRNA transcripts by in-situ hybridization (ISH). Surprisingly, we found that Prx-expressing myelinating Schwann cells were a source of IL-34 (Fig. 5l). In the absence of IL-34, PNS macrophages in the SN showed increased surface marker expression of CD45 and CD11b, suggesting a shift in phenotype further away from microglial characteristics (Fig. 5m). Taken together, these results reveal shared and unique developmental programs, including instructive cytokines, between PNS macrophages and CNS microglia.

**Nerve environment shapes PNS macrophage signature**. To investigate the extent to which PNS macrophage identity is specified by ontogeny or nerve environment, we individually sorted YFP− embryonic-derived and YFP+ HSC-derived PNS macrophages from SNs of Flt3Cre LSL-YFP$^{fl/fl}$ mice and performed single-cell RNA-seq (Fig. 6a). We captured a total of 935 YFP− cells and 3186 YFP+ cells. Unsupervised clustering analysis of all 4121 cells revealed 5 separate clusters, with the majority of PNS macrophages belonging to one major group (clusters 1, 2, and 3) and the remainder falling into 2 smaller clusters (4 and 5) (Fig. 6b and Supplementary Data 5).

Although we observed a significant overlap of YFP− and YFP+ macrophages in clusters 1, 3, 4, and 5, cluster 2 contained mostly YFP+ macrophages (Fig. 6b). Interestingly, cluster 2 was defined by *Ccr2* expression, which is consistent with this subset arising from circulating precursors (Fig. 6c). Indeed, we confirmed by flow cytometry that CCR2+ PNS macrophages were only found in the YFP+ fraction (Fig. 6f and Supplementary Fig. 14). We also observed varying heterogeneity between the overlapping clusters (Fig. 6c). For instance, cluster 5 was easily distinguished by proliferation genes *Mki67* and *Top2a*. Cluster 4 was also relatively distinct and showed enrichment for *Ly6e*, *Ninj1*, *Retnla*, and *Wfdc17*, potentially representing a previously undescribed activation state. Cluster 3 selectively expressed early activation genes *Fos*, *Jun*, and *Egr1*. Cluster 1 was slightly enriched for *Lyve1* expression compared to cluster 2 (Fig. 6c). We confirmed Lyve1 expression in a subset of PNS macrophages by immunostaining (Fig. 6g). Interestingly, we also saw axonal expression of YFP in Flt3-Cre LSL-YFP mice (Fig. 6g), which is in accordance with previous findings that Flt3 is expressed in neurons and may play a role in neural stem cell proliferation and survival[39].

Despite the identification of separate clusters in our data, we observed no obvious difference in the expression of PNS macrophage-enriched or microglial activation-associated transcripts between the main clusters. Specifically, *Apoe*, *Cp*, *H2-Aa*, *Cd74*, *Ms4a6c*, *Ifitm3*, *Anxa5*, *Cybb*, and *Cd52* were nearly identical between clusters 1, 2, and 3 (Fig. 6d). We also observed no difference in *Cx3cr1* and *Trem2* between these clusters. Importantly, all of these genes were similarly expressed between YFP− and YFP+ macrophages (Fig. 6e). We conclude that embryonic- and HSC-derived PNS macrophages are transcriptionally similar and that the nerve environment confers a predominant effect over developmental origin on PNS macrophage identity.

**Discussion**

Here we characterized the transcriptomes of PNS resident macrophages across various nerve types spanning both axons and cell bodies and innervating a wide range of somatic targets. We show that self-maintaining PNS macrophages lack expression of the master regulator Sall1 conferring CNS microglial identity, but nonetheless express a substantial number of genes that define homeostatic and activated microglia. Similar to microglia, PNS macrophages were also partially dependent on IL-34. The

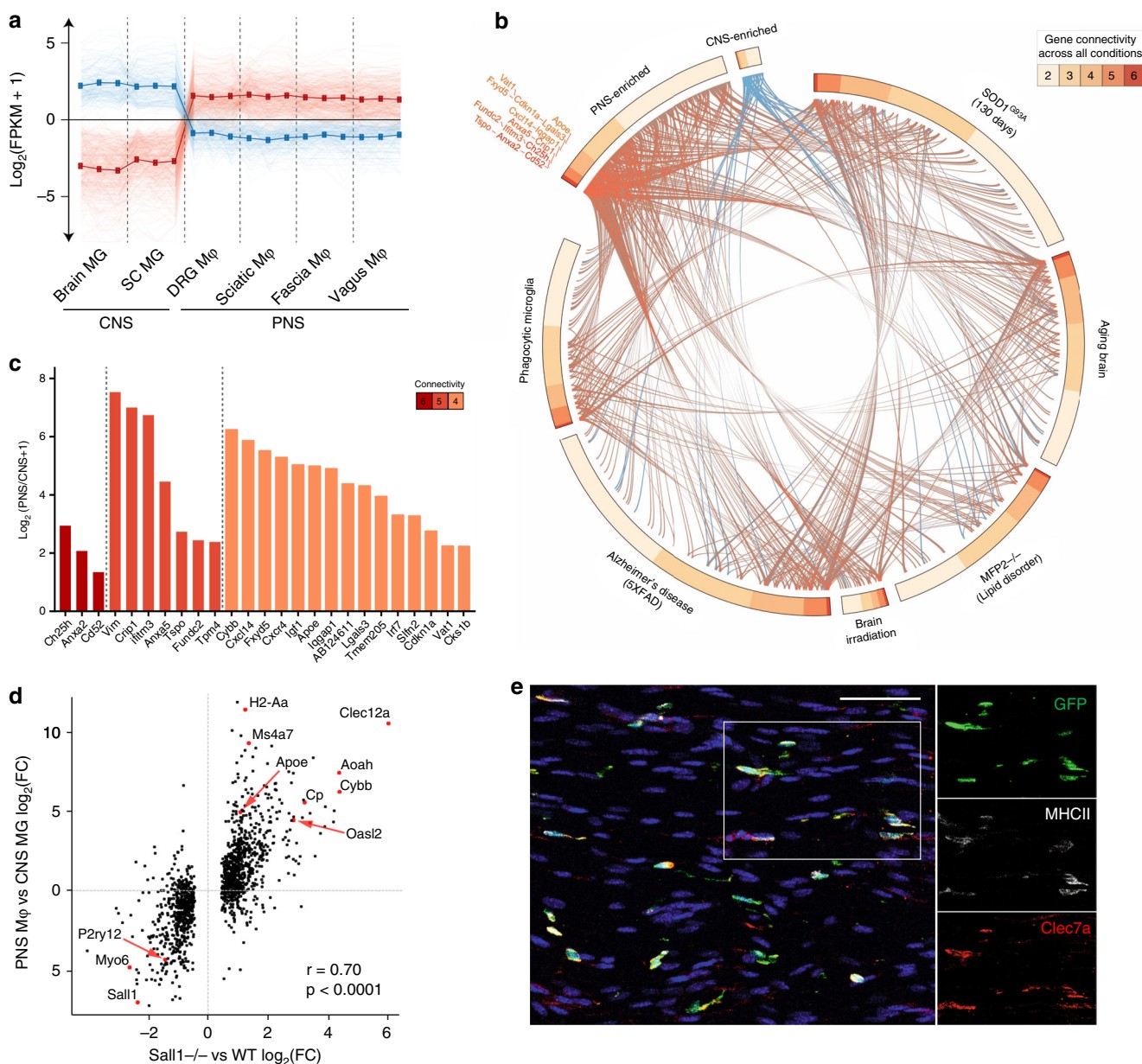

**Fig. 3 PNS macrophages constitutively express transcripts associated with activated microglia. a** Expression pattern of DEGs defined as PNS-enriched (red) or CNS-enriched (blue). CNS microglia includes brain and spinal cord and PNS macrophages include DRG, vagal, fascial, and sciatic nerves. **b** Circos plot showing the number of connections (gene connectivity) between GSEA-scored genes from microglia in 6 neurodegenerative and aging-associated conditions as defined in Krasemann et al.[6] and neural macrophage-enriched genes from either PNS macrophages (red) or CNS microglia (blue). **c** PNS-enriched genes from connectivity groups 6–4 expressed as $\mathrm{Log}_2$ fold change (PNS macrophages/CNS microglia). **d** Expression plot comparing PNS macrophage-enriched genes (expressed as PNS macrophage/CNS microglia Log2FC) and Sall1$^{-/-}$ microglia-enriched genes (expressed as Sall1$^{-/-}$/wild-type microglia Log2FC) from Buttgereit et al.[28]; *r*, correlation coefficient; *p*, *p*-value for linear regression analysis. **e** Representative immunohistochemistry in sciatic nerves of CX3CR1$^{\mathrm{GFP/+}}$ mice showing DAPI (blue), GFP (green), MHCII (white), and Clec7a (red). Scale bar, 50 μm. Source data are available as a Source Data file.

convergence of both HSC-derived and embryonic PNS macrophages into a single population at steady state underscores the importance of tissue environment for specifying PNS macrophage identity.

Our findings demonstrating that PNS macrophages share a significant overlap in gene expression with CNS microglia are in line with recent findings that show microglial genes being expressed in nerve-associated macrophages in gut, skin, and adipose tissues[17–20]. Given that our study focuses on PNS macrophages residing in the nerve proper whereas these studies examined macrophages positioned in close proximity to nerves, including nerve termini, it seems

likely that common tissue-derived factors may induce nerve-imprinted signatures in macrophages along the entire neuron. One such factor may be transforming growth factor-β (TGF-β). TGF-β signaling is vital for the expression of homeostatic genes in microglia[2,5]. The higher expression of *Tgfbr1* in PNS macrophages and CNS microglia at steady state supports the role of TGF-β signaling in neural resident macrophages.

Although common instructive signals may exist, *Sall1* expression was very low to absent in PNS macrophages. This likely represents a key difference in PNS and CNS nerve environments. The identification of additional genes that were purely expressed

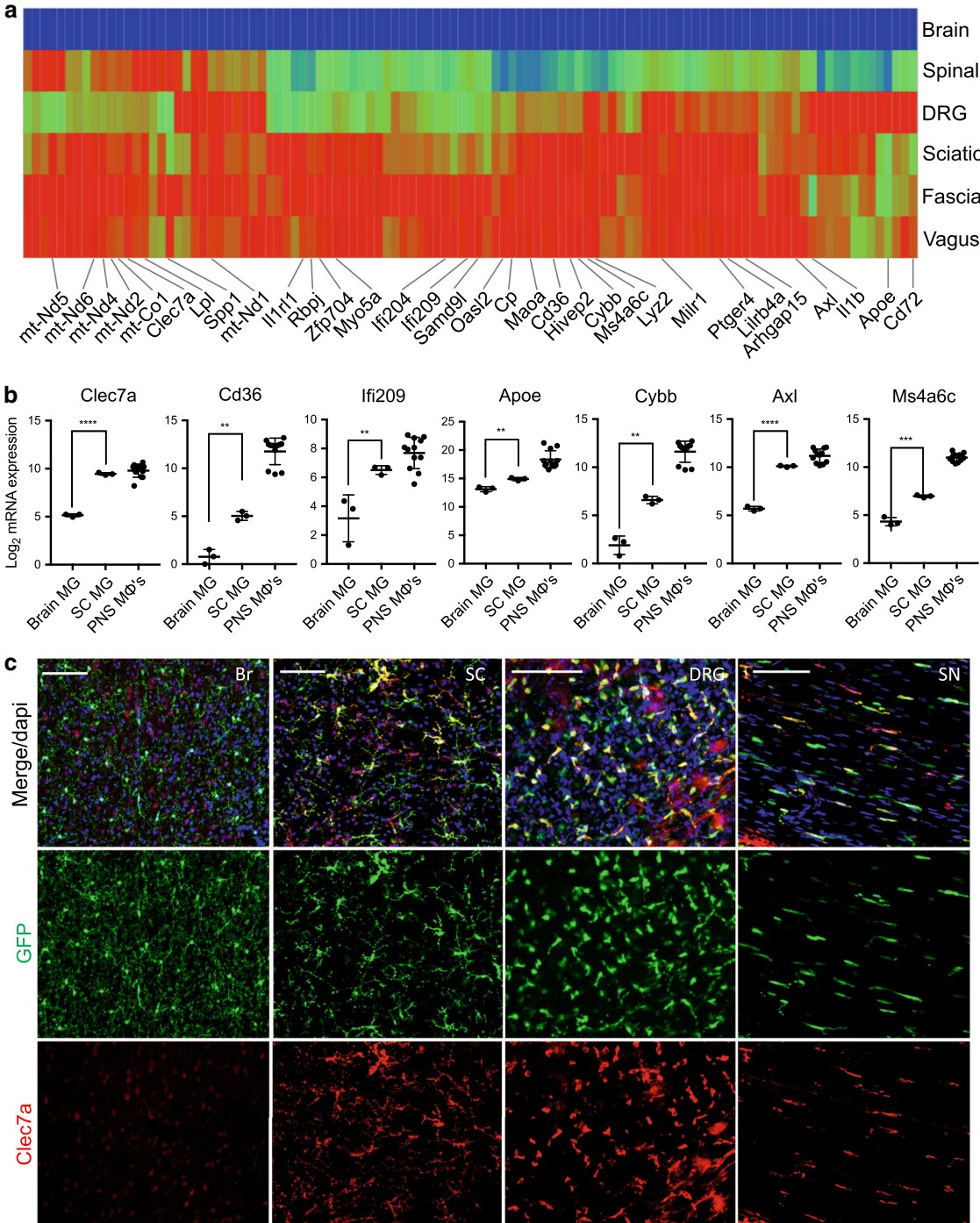

**Fig. 4 Zonation of PNS macrophage-enriched transcripts in neural resident macrophages. a** Heat map showing genes corresponding to PNS to CNS zonation pattern (PNS macrophages high, spinal cord microglia high/intermediate, brain microglia low). Each box represents average from three replicates. **b** Gene expression analysis of individual genes following PNS to CNS zonation pattern. Each dot represents one replicate. Unpaired *t*-test. Data are mean ± SD. *P < 0.05, **P < 0.01, ***P < 0.001, ****P < 0.0001. **c** Representative Clec7a staining in brain (Br), spinal cord (SC), dorsal root ganglia (DRG), and sciatic nerve (SN) of CX3CR1$^{GFP/+}$ mice. Scale bars are 100 μm. Source data are available as a Source Data file.

in CNS microglia may provide clues for the basis of this difference. Future studies should address genetic and environmental factors that regulate *Sall1* expression and examine whether it is expressed in a context-dependent manner in PNS macrophages.

The constitutive expression of a broad set of microglial activation genes in PNS macrophages and, to a lesser degree, in spinal cord microglia suggests that such programs may be intrinsic to their functioning in distinct neuronal environments. Indeed, it

was found that sympathetic nerve-associated macrophages and macrophages at CNS interfaces also appear to be constitutively activated[17,40,41]. In addition, an overlap in many of the same activation genes are induced in phagocytic microglia localized to white matter tracts during brain development[42]. In addition to their role in phagocytic functions, activation-associated genes may also play a role in pathogen response. This is supported by the recent finding that nerve-associated macrophages residing in

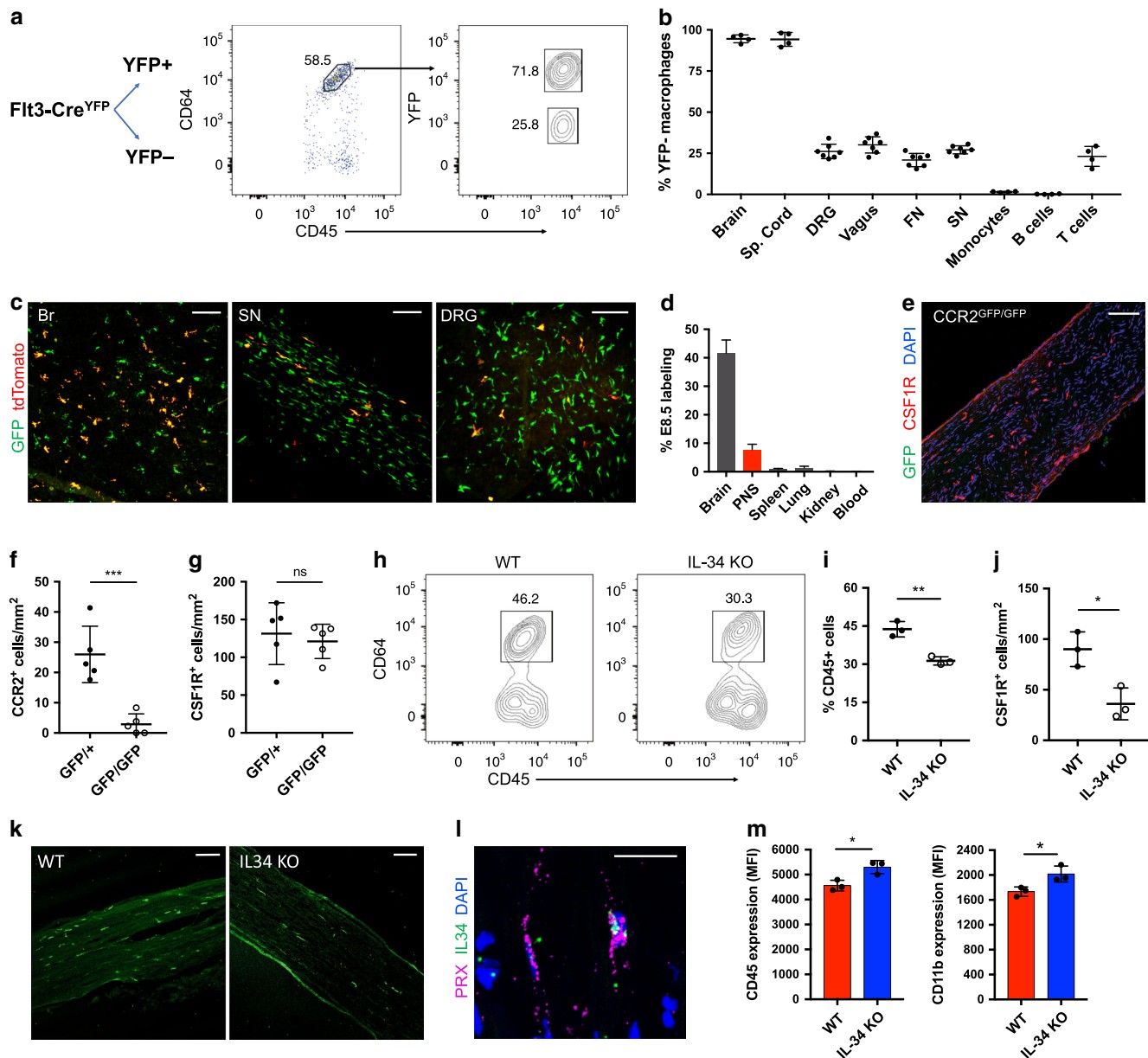

**Fig. 5 Shared and distinct developmental programs in PNS macrophages. a** Schematic and flow cytometric gating for analysis of YFP expression in PNS macrophages from Flt3-Cre LSL-YFP sciatic nerves. **b** Comparison of YFP− macrophage ratios (expressed as % YFP− cells of population) across neural resident macrophage populations, monocytes, B cells, and T cells ($n = 4-7$ mice per group). **c** E8.5 tdTomato labeling and GFP expression in brain (left panel), sciatic nerve (middle panel), and DRG (right panel) of CSF1R^Mer-iCre-Mer × tdTomato^fl/fl × CX3CR1^GFP/+ mice. Scale bar, 100 μm. **d** Quantification of E8.5 tdTomato labeling in brain, PNS, spleen, lungs, kidney, and blood from newborn pups ($n = 2$ mice for blood analysis, $n = 3$ mice for tissue analysis). **e** Histological representation of PNS macrophages in CCR2^GFP/GFP sciatic nerves. Scale bar, 100 μm. **f** Quantification of CCR2+ CSF1R+ macrophages in sciatic nerves in CCR2^GFP/+ and CCR2^GFP/GFP mice ($n = 5$ mice per group). **g** Quantification of total CSF1R+ macrophages in CCR2^GFP/+ and CCR2^GFP/GFP mice ($n = 5$ mice per group). **h** Flow cytometric gating and (**i**) analysis of sciatic nerve macrophages from wild type (WT) and IL-34 knockout (KO) mice ($n = 3$ mice per group). **j** Quantification of sciatic nerve macrophages in WT and IL-34 KO mice ($n = 3$ mice per group). Scale bar, 100 μm. **k** Representative imaging of WT and IL-34 KO sciatic nerves. CSF1R staining in green. Scale bar, 120 μm. **l** In-situ hybridization in WT sciatic nerve showing IL-34 colocalization with Prx-expressing Schwann cell. Scale bar, 50 μm. **m** CD11b and CD45 expression in PNS macrophages from WT (red) and IL-34^LacZ/LacZ (blue) mice. For imaging quantification, at least three images were taken and analyzed from each tissue and averages from each group were plotted. Each dot represents one mouse. All data are mean ± SD. Data comparing wild type and knockouts analyzed by unpaired t-test. *$P < 0.05$, **$P < 0.01$, ***$P < 0.001$, ****$P < 0.0001$; NS, not significant. Source data are available as a Source Data file.

lungs readily respond to viral infection[21]. Further studies are needed to determine whether immune response to pathogens or phagocytic and tissue remodeling programs are more prevalent at steady state in PNS macrophages.

Although the connection between neurodegeneration and immune response has become increasingly apparent, how distinct expression patterns relate to neuronal disease, development, and homeostasis remains a mystery. Ontogeny seems to be an important factor, as shown by several studies that found upregulation of disease-associated genes and neurotoxic functions in transplanted cells with hematopoietic origin[32,33]. Our results show that both embryonic and HSC-derived PNS macrophages

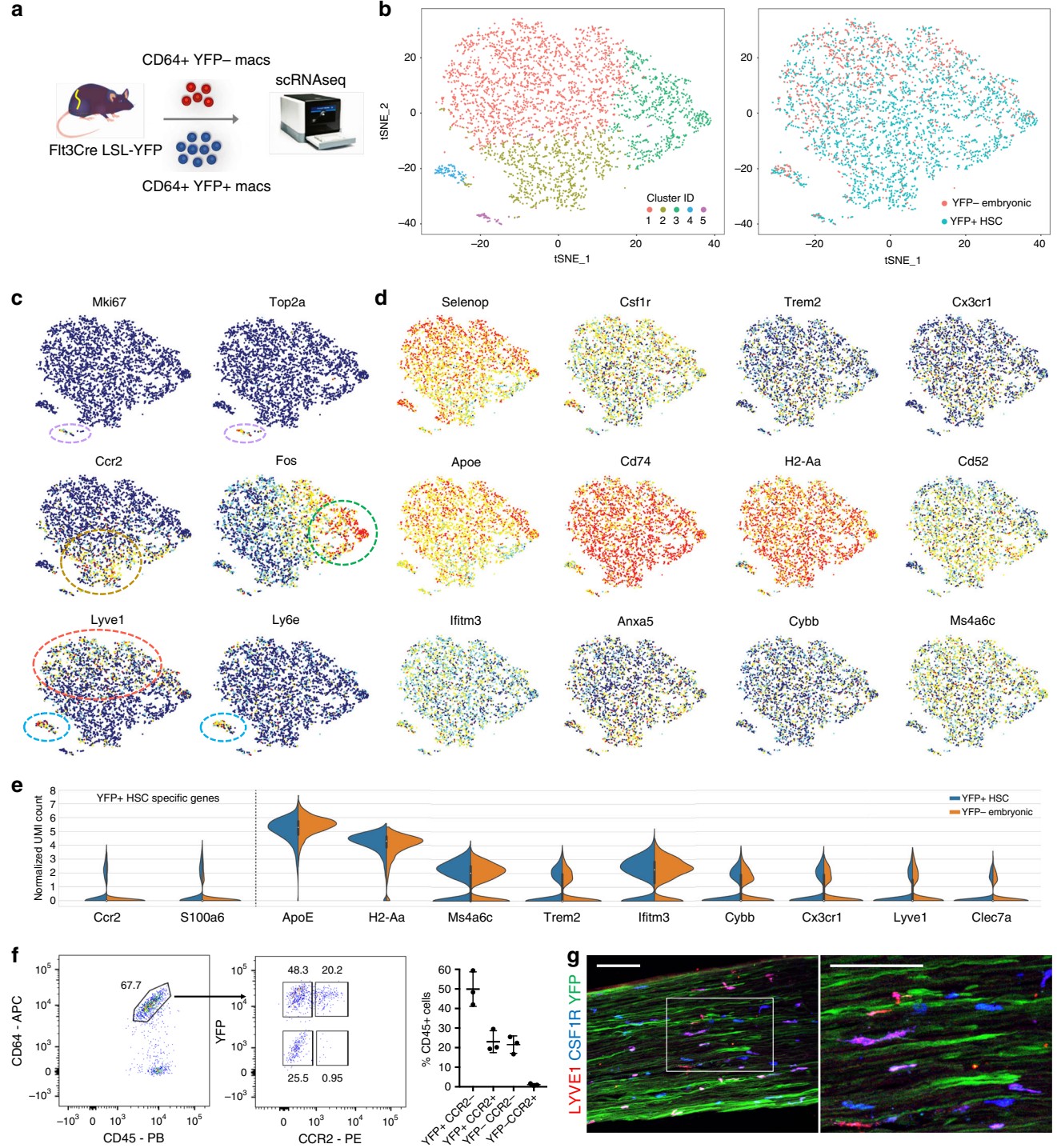

**Fig. 6 Nerve environment shapes transcriptional identity of PNS macrophages. a** Schematic for isolation and separate single-cell RNA sequencing of YFP− embryonic and YFP+ HSC-derived macrophages from sciatic nerves of Flt3-Cre LSL-YFP$^{fl/fl}$ mice. Two single-cell libraries from YFP− and YFP+ macrophages were prepared using the 10× single-cell RNA-seq platform (Chromium Controller image provided by 10× Genomics. **b** t-SNE plot of 4121 CD64+ CD45int cells from pooled sciatic nerves ($n = 18$) showing unsupervised clustering (left panel) and overlay of YFP− (red) and YFP+ (blue) populations (right panel). **c, d** t-SNE plots depicting distribution of (**c**) transcripts relating to clusters in b and (**d**) transcripts associated with microglial activation across combined embryonic and HSC-derived PNS macrophages. **e** Violin plots of marker gene expression in YFP+ HSC-derived and YFP− embryonic groups. **f** Flow cytometric identification of CCR2+ macrophages in a subset of YFP+ macrophages. Gating is representative of at least seven nerves examined over three separate experiments. **g** Representative imaging of a subset of Lyve1+ macrophages in Flt3-Cre LSL-YFP$^{fl/fl}$ mouse sciatic nerve. Scale bar, 100 μm. Source data are available as a Source Data file.

exist as transcriptionally similar populations in adult nerves regardless of origin, thus suggesting a more important role for nerve environment than ontogeny in programming neural resident macrophages at specific sites. Moreover, as PNS macrophages are a naturally occurring population rather than a population recruited in response to injury or pathology, they serve as a useful standard of comparison in the context of normal physiological function.

Up to now, the brain and spinal cord of the CNS are the only tissues thought to possess microglia. Our comparison of resident macrophages from both PNS and CNS challenges this notion and supports the idea that some microglial features and transcriptional programs are shared in macrophages across the nervous system, whereas critical features remain distinct. The increasing amount of transcriptomic data on microglia and nerve-associated macrophages from different tissues and across organs presents opportunities for novel discoveries that may be applied to neurodegenerative conditions and beyond.

## Methods

**Experimental animals**. Mouse care and experiments were performed in accordance with protocols approved by the Institutional Animal Care and Use Committee at Washington University in St. Louis under the protocols 20170154 and 20170030. Mice were kept on a 12 h light–dark cycle and received food and water ad libitum. The following strains were used: C57/B6 CD45.1 (stock number 002014), CX3CR1[GFP/+] (Cx3cr1tm1Litt/LittJ; stock number 008451), LysM[cre/+] (stock number 004781), and Rosa[Lsl-Tomato] (stock number 007905) mice were purchased from Jackson Laboratory (JAX) and bred at Washington University. CSF1R[Mer-iCre-Mer] mice[43] were obtained from JAX (stock number 019098) and backcrossed fully to C57BL/6 background using the speed congenics core facility at Washington University. MPZ-Cre mice, previously described[44], were crossed with CX3CR1[GFP/+] mice. Flt3-Cre LSL-YFP[fl/fl][35] mice were kindly provided by D. Denardo, CCR2[gfp/+] (B6(C)-Ccr2tm1.1Cln/J, (JAX stock number 027619) mice were kindly provided by K. Lavine, IL34[LacZ/LacZ] mice were previously described and generated at Washington University[15], and Sall1-[GFP] mice[28,45] were kindly provided by M. Rauchman.

**Immunohistochemistry**. For whole-mount imaging, samples were harvested and immediately stored in 4% paraformaldehyde (PFA) containing 40% sucrose overnight. Samples were then washed in phosphate-buffered saline (PBS), blocked, stained, and imaged. For frozen sections, samples were collected, stored in PFA/sucrose, and embedded into optimal cutting temperature compound. Fifteen-micrometer cuts were made for SNs. Sections were then blocked in 1% bovine serum albumin, stained, and imaged. Antibodies to the following proteins were used: anti-GFP, Clec7a (Clone R1-8g7), CSF1R (R&D Systems, accession number P09581), MHCII (IA/IE clone M5/114.15.2), and LYVE1 (ab14917).

**Preparation of single-cell suspensions**. For blood, mice were bled from the cheek immediately before killing and cells were prepared as previously described. For nerves and all other tissues, mice were killed and perfused with PBS. Nerves were collected and kept on ice until dissociation. For ImmGen samples, nerves from 4 to 20 mice were pooled for each replicate. Cells were then incubated with gentle shaking for 20 min in digestion media containing collagenase IV, hyaluronidase, and DNAse. Cells were then washed and filtered through 70 μm cell strainers. For brain and spinal cord, myelin was removed using a 40/80% Percoll gradient.

**Flow cytometry**. Single-cell suspensions were stained at 4 °C. Dead cells were excluded by propidium iodide (PI). Antibodies to the following proteins were used: B220 (clone RA3-6B2), CCR2 (clone SA203G11), CD3e (clone 145-2C11), CD4 (clone RM4-5), CD8 (clone 53-6.7), CD11b (clone M1/70), CD16 (clone 2.4G2), CD45 (clone 30-F11), CD64 (clone X54-5/7.1), CD115 (clone AFS98), GR1 (clone 1A8), and Ly6C (clone HK1.4). Cells were analyzed on a LSRII flow cytometer (Becton Dickinson) and analyzed with FlowJo software.

**Cell sorting**. For bulk RNA-seq, cells from 6-week-old male mice were double sorted on a FACSAria II (Becton Dickinson) for a final count of 1000 cells into lysis buffer according to the ImmGen Consortium standard operating protocol. Tissues were collected into culture medium on ice and subsequently digested with collagenase IV, hyaluronidase, and DNAse. Following digestion, samples were washed and kept on ice until sorting. The sort was repeated so that all macrophages were sorted twice, with a minimum of 1000 cells recovered. During the second round, cells were sorted directly into 5 μl TCL buffer (Qiagen) containing 5% beta-mercaptoethanol. Samples were kept at −80 °C until further processing. For sorting in preparation of the Flt3-Cre LSL-YFP single-cell Seq experiments, SNs from 19

male mice aged 10–12 weeks were combined and CD64+ CD45int macrophages were sorted individually into YFP+ and YFP− groups, yielding 18,000 and 5,000 cells, respectively. Both groups were immediately run on the 10× Genomics Chromium Controller according to the manufacturer's protocol.

**Parabiosis**. Parabiotic pairs were generated as previously described[46]. C57/B6 (CD45.1) mice were paired with Lyz2Cre tdTomato (CD45.2) mice. Mice were injected with buprenorphine-SR subcutaneously prior to surgery. After 10 weeks, mice were killed and nerve tissue was examined by flow cytometry and imaging to detect hematopoietic contribution to PNS resident macrophages. T cells in blood was used as a positive control.

**Pulse chase**. Male and female heterozygous CSF1R[Mer-iCre-Mer] tdTomato were fed tamoxifen diet for 4 weeks to label resident cells. Blood was collected at day 0, 3 weeks, 4 weeks, and 8 weeks after tamoxifen removal. Peripheral nerves from SNs, FNs, VNs, and DRG were pooled (PNS) and examined along with pooled brain and spinal cord (CNS) by flow cytometry at day 0 and 8 weeks following tamoxifen removal.

**RNA-seq and data analyses**. Library preparation, RNA-seq, data generation and quality-control was conducted by the ImmGen Consortium according to the consortium's standard protocols (https://www.immgen.org/Protocols/ImmGenULI_RNAseq_methods.pdf). In short, the reads were aligned to the mouse genome GRCm38/mm10 primary assembly and gene annotation vM16 using STAR 2.5.4a. The raw counts were generated by using featureCounts (http://subread.sourceforge.net/). Normalization was performed using the DESeq2 package from Bioconductor. Differential gene expression analysis was performed using edgeR 3.20.9 in a pairwise manner among all conditions, and a total of 12,240 DEGs were defined with a $p$-value ≤ 0.001 and ≥4-fold difference. To construct the correlation plot, Euclidean distance among samples were calculated based on the differential expression matrix and clustering was performed using the ward.D2 algorithm in R. CNS/PNS shared, PNS-specific, and CNS-specific genes were determined by subclustering the DEGs based on the expression pattern with a refined $k$-mean clustering using R followed by manual curations. For neuronal microenvironment analysis, only the transcriptome profile of macrophages and microglia from PNS and CNS were used and analyzed through the same pipeline as mentioned above.

**Comparison of published microglia data**. External datasets for circos plot were obtained from Krasemann et al.[6]. Genes that are enriched in SOD1G93A, aging brain, MFP2[−/−], brain irradiation, AD (5XFAD), and phagocytic microglia conditions were previously generated in Krasemann et al.[6]. By comparing the PNS- and CNS-enriched genes with the disease signatures, we were able to define the number of conditions shared by the genes and coined the term as "connectivity". Only genes with connectivity of two or above are shown in the circos plot. For the comparison with Sall1[−/−] data set[24], the log₂ fold change of genes between CNS microglia and PNS macrophages were calculated and compared against the public data set. Correlation coefficient and $p$-value were calculated by lineregress in Scipy using Python3.

**Embryonic labeling**. Homozygous CX3CR1-GFP female mice were rotated daily with CSF1R[Mer-iCre-Mer] tdTomato male mice and checked for plugs in the morning. Plug-positive females were administered 1.5 mg of 4-Hydroxytamoxifen (Sigma, catalg number H6278) and 1 mg of progesterone (Sigma, catalog number P0130) dissolved in corn oil by oral gavage 8 days following identification of plugs to pulse the embryos at embryonic 8.5 days. Following birth, pups were immediately sacrificed. Blood was collected for flow cytometric analysis and tissues were fixed in PFA/sucrose for imaging.

**In-situ hybridization**. RNA ISH was performed using the ViewRNA Tissue Assay Core Kit (Invitrogen, catalog number 19931) according to the manufacturer's instructions, with a probe set designed for IL-34 (VB1-14592-VT) and Prx (VB6-3201318-VT). Probe sets were purchased from ThermoFisher.

**Cell preparation, 10× single-cell library preparation, sequencing, and analyses**. A total of 4500 Flt3-negative and 16,000 Flt3-positive cells were loaded to separate lanes of the 10X Chip for preparation of two single-cell libraries. The library preparation was performed according to the manufacturer's instructions (Chromium Single-cell v2; 10× Genomics, USA). A total of 153 M and 174 M reads were sequenced for Flt3-negative and Flt3-positive libraries, respectively, using Illumina HiSeq2500. Reads were mapped by using the cellranger pipeline v2.1.1 onto the reference genome grcm38/mm10. We filtered cells for those with ≥50,000 mapped reads, leaving ~1k Flt3-negative and 4k Flt3-positive cells. Downstream analyses were performed by using the package Seurat2 in R.

**Reporting summary**. Further information on research design is available in the Nature Research Reporting Summary linked to this article.

## Data availability

The bulk and single-cell RNA-seq data set is available in Supplementary Data 1–5 and online in Gene Expression Omnibus (GEO) database (GSE122108 [https://www.ncbi.nlm.nih.gov/geo/query/acc.cgi?acc=GSE122108] and GSE146510 [https://www.ncbi.nlm.nih.gov/geo/query/acc.cgi?acc=GSE146510]). All other datasets generated and analyzed in the current study are provided in the source data. Additional data that support the findings of this study are available from the corresponding author upon reasonable request. Source data underlying Figs. 1–6 are available as a Source Data file.

## Code availability

The source code for data analysis and visualization in this study can be found at https://github.com/aldrinyim/PNS-macrophages.

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

## Acknowledgements

We thank all our collaborators at Washington University School of Medicine (WUSM) for their advice and discussion. We thank Richard Head, Ruteja Barve, and the Genome Technology Access Center (GTAC) at WUSM for discussion and assistance with RNA sequencing. We thank John Baer and David DeNardo for providing Flt3-Cre LSL-YFP^fl/fl mice. We thank Kory Lavine for providing Ccr2^GFP/GFP mice and Marco Colonna and Susan Gilfillan for providing IL-34^LacZ/LacZ mice. We thank Jesse Williams, Amy Strickland, and Yingyue Zhou for experimental assistance. We thank Wilbur Song for discussion and Joseph Bloom for reading the manuscript. We also thank our colleagues at the ImmGen Project consortium, including Christophe Benoist PI of the Immgen Project), fellow scientists who submitted samples, and Kumba Seddu for coordinating transfer of samples and data. We thank Nan Zhang for help with revisions and Emma Erlich for help with data analysis. This project was supported by NIH grants R37AI049653 and DP1DK109668 to G.J.R., RF1AG013730 and R01NS105645 to J.M., and the Principles in Pulmonary Research training grant (T32 HL007317-41) to P.L.W. Further support was provided by P30AR073752 that supports Rheumatic Diseases Research Resource-Based Center and NIH R24 AI072073 that funds the ImmGen Project.

## Author contributions

P.L.W. purified macrophage populations, designed, and performed the experiments, analyzed data, and wrote the manuscript. A.Y. analyzed RNA-seq data and helped with

experiments and writing of the manuscript. K.K. purified macrophage populations and discussed results. D.A. and R.C. helped with experiments and discussed results. M.C. provided conceptual feedback and the IL-34$^{-/-}$ strain. G.J.R. and J.M. designed and supervised the experiments, acquired funding, and edited the manuscript. The Immunological Genome Project set standards for data acquisition, conducted sequencing, QC, and generation of raw data for bulk RNA samples.

## Competing interests

The authors declare no competing interests.
