## [Peer Review File · Nature Communications]

This manuscript has been previously reviewed at another journal that is not operating a transparent peer review scheme. This document only contains reviewer comments and rebuttal letters for versions considered at Nature Communications .

Reviewers' comments:

Reviewer #1 (Remarks to the Author):

The authors have very nicely addressed the comments that were raised. I fully support acceptance of the manuscript.

Reviewer #4 (Remarks to the Author):

In this article, Wang and colleagues characterized the transcriptional identity of macrophages isolated from the DRG and peripheral nerves of naïve mice and further compared them with steady-state CNS microglia. Using multiple approaches such as parabiosis, fate mapping and RNA-seq, they further confirmed that heterogeneous PNS macrophages were derived from both embryonic and hematopoietic precursors. Interestingly, although a subset of cells were likely recruited from circulation, PNS macrophages were largely self-maintained.

Despite considerable evidence to support the contribution to nerve regeneration and neuropathic pain, PNS macrophages have not been fully profiled. Therefore, the findings in this study may have potential to significantly expand our current understanding of neuroinflammation implicated in many neurological disorders. However, I have serious concerns about the major conclusions of this study that the authors need to address before publication.

1. The evidence presented in this manuscript sufficiently supports that PNS macrophages are quite different from CNS microglia. Phenotypically, PNS macrophages are *Sall1*-*CCR2*^{+/-}, whereas microglia are *Sall1*⁺*CCR2*⁻. Given that microglia are “resident macrophages” in the CNS, I do not see the novelty to reclaim macrophages in the PNS as “microglia-like”. Although the authors have agreed to “tone down the claim”, in their response to Reviewer #2, I have not noticed many changes made in this revised manuscript.
2. The authors agree that DRG macrophages and nerve-associated macrophages “were significantly different” (Page 6/line 127) by RNA-seq analysis. There are also some important biological differences between them, particularly in the context of nerve injury. For example, rapidly accumulated macrophages at the nerve injury site predominantly originate from circulation. In contrast, resident cell proliferation contributes to both microgliosis and DRG macrophage expansion (Gu et al., Cell report 2016; Huang et al., Nat Neurosci 2018; Yu et al., Nat Commun 2020). Furthermore, both microglia and DRG macrophages can crosstalk with surrounding neurons, a hallmark of neuroinflammation. Therefore, scRNA-seq on a few sciatic nerve macrophages is not representative in investigating the extent to which microenvironment can shape PNS macrophage transcriptome signature. Instead, the authors should perform scRNA-seq on DRG macrophages of *Flt3Cre* LSL-*YFPfl/fl* mice to provide more insights for PNS macrophage biology.
3. The authors favor the view that CNS microglia “depend on IL-34 for development” (Wang et al., Nat Immunol, 2012). IL-34 was further suggested as a bridge between CNS microglia and PNS macrophages during development. Arguing against this, Greter and colleagues found that “IL-34 did not control the embryonic development of microglia but contributed to microglia homeostasis in the adult” (Immunity, 2012). Notably, in Wang’s paper, adult microglia were reduced but not abolished in IL-34 KO mice. This discrepancy in the literature should be considered and discussed.
4. It is somewhat unexpected and potentially important that macrophages in sciatic nerves of IL-34^{-/-}

mice were reduced compared to WT mice, in part likely due to lack of IL-34 expression in Schwann cells (Fig. 5). Is IL-34 also expressed in the DRG? Are DRG macrophages impacted by the loss of IL-34? Does CSF1, another cognate CSF1R ligand, compensate for the loss of IL-34? Notably, CSF1 is detected in Schwann cells of sciatic nerve (Trias et al., *Glia* 2019), constitutively expressed in both satellite cells and sensory neurons in the DRG (Guan et al., *Nat Neurosci* 2016).

5. What is “subcutaneous fascial nerve” described in this study? There are sensory nerve fibers present in the fascia layer. But I am not aware there is a distinct large nerve which was illustrated in Fig. 1a. Please provide explanation and references.

6. Flt3-Cre-LSL-YFPfl/fl mice were used for scRNA-seq in Fig. 6. Surprisingly, myelin or neurofilament in sciatic nerve was positive for YFP (Fig. 6g). No additional information was provided in the manuscript to suggest a different mouse line was used in Fig. 6g. Therefore, I will have a serious concern about the specificity of YFP+ myeloid cells sorted for scRNA-seq. A minor point, please amend figure legends which only describes 6 out of 7 panels in the figure.

Minor points:

1. The order of some references and figures listed in the text is often confusing. The following changes are suggested for improvement:

- a. Page 3/line 48, “during embryogenesis (14) and depend on IL-34 for development (15)”;
- b. Page 4/line 61-62, “CX3CR1-GFP+ cells were located in the endoneurium (Fig 1b) and expressed colony-stimulating factor 1 receptor (CSF1R), also known as CD115 (Fig. 1c).

2. Figs. 1a-c.

- a. In Fig. 1b, Col1 staining was used to mark the perineurial layer of sciatic nerve. But Col1 was not introduced in the manuscript.
- b. In the enlargement of the outlined area in Fig. 1c, one CX3CR1-GFP+ cell in the center did not coexpress CSF1R. Again this makes me wonder how specific the sorted cells were.
- c. To better illustrate macrophages, may change MPZ staining from red to blue.
- d. Please describe CX3CR1GFP/+MPZ tdTomato mice in methods.
- e. Fig. 1h, please also provide representative images from CD45.2+ tdTomato+ parabiont.
- f. Fig. 1i: % tdTomato+ cells should be plotted individually based on the tissue sources.

3. Fig. 3e, I do not quite understand the rationale to use Clec7a and MHCII staining to rule out the technical artifact in macrophages signature. A MHCIIhi (Lyve1lo) tissue resident macrophage population was already found surrounding the nerves at steady state (Chakarov et al., *Science* 2019). Therefore, high MHCII expression is neither surprising nor activated microglia-specific.

4. Fig. 5a: Please include FACS plot of CNS microglia as a gating control.

5. Page 10/line 210: “GFP+ cells were totally absent from Ccr2GFP/GFP mice (Fig. 5e, f)”. However, CCR2+ cells were still detected in 3 out of 5 Ccr2GFP/GFP mice by immunostaining (Fig. 5f). Image quantitation needs to be better explained in Methods.

6. IL34 KO mice can be used as a negative control for ISH staining. Please also provide the resource of ISH probes in Methods.

7. In the future, the authors may consider including female mice and expanding their effort in the context of nerve injury.

Reviewer #4 (Remarks to the Author):

In this article, Wang and colleagues characterized the transcriptional identity of macrophages isolated from the DRG and peripheral nerves of naïve mice and further compared them with steady-state CNS microglia. Using multiple approaches such as parabiosis, fate mapping and RNA-seq, they further confirmed that heterogeneous PNS macrophages were derived from both embryonic and hematopoietic precursors. Interestingly, although a subset of cells were likely recruited from circulation, PNS macrophages were largely self-maintained.

Despite considerable evidence to support the contribution to nerve regeneration and neuropathic pain, PNS macrophages have not been fully profiled. Therefore, the findings in this study may have potential to significantly expand our current understanding of neuroinflammation implicated in many neurological disorders. However, I have serious concerns about the major conclusions of this study that the authors need to address before publication.

We thank the reviewer for careful review of the manuscript and expert comments. We have revised the manuscript accordingly and are grateful for the feedback that has helped us improve this body of work.

1. The evidence presented in this manuscript sufficiently supports that PNS macrophages are quite different from CNS microglia. Phenotypically, PNS macrophages are Sall1-CCR2+/-, whereas microglia are Sall1+CCR2-. Given that microglia are “resident macrophages” in the CNS, I do not see the novelty to reclaim macrophages in the PNS as “microglia-like”. Although the authors have agreed to “tone down the claim”, in their response to Reviewer #2, I have not noticed many changes made in this revised manuscript.

The reviewer makes a good point. The main feature of microglia that we believe merits a strong highlight is that a set of genes considered those defining activated microglia are constitutively expressed by the various PNS macrophages. We have now carefully combed through the manuscript and edited it, including the title, to highlight that particular similarity, while toning down other text that may have implied greater similarity than warranted. Please see highlighted text changes throughout the manuscript that illustrate the numerous places in which we changed the tone of the text.

2. The authors agree that DRG macrophages and nerve-associated macrophages “were significantly different” (Page 6/line 127) by RNA-seq analysis. There are also some important biological differences between them, particularly in the context of nerve injury. For example, rapidly accumulated macrophages at the nerve injury site predominantly originate from circulation. In contrast, resident cell proliferation contributes to both microgliosis and DRG macrophage expansion (Gu et al., Cell report 2016; Huang et al., Nat Neurosci 2018; Yu et al., Nat Commun 2020). Furthermore, both microglia and DRG macrophages can crosstalk with surrounding neurons, a hallmark of neuroinflammation. Therefore, scRNA-seq on a few sciatic nerve macrophages is not representative in investigating the extent to which microenvironment can shape PNS macrophage transcriptome signature. Instead, the authors should perform scRNA-seq on DRG macrophages of Flt3Cre LSL-YFPfl/fl mice to provide more insights for PNS macrophage biology.

In agreement with the editor’s comments, we argue that this request is beyond our scope for the present study. We have already made the conclusion that DRG macrophages are different. It is in light of

these differences that our findings may be of interest and future work can further address these differences in the context of nerve injury.

3. The authors favor the view that CNS microglia “depend on IL-34 for development” (Wang et al., *Nat Immunol*, 2012). IL-34 was further suggested as a bridge between CNS microglia and PNS macrophages during development. Arguing against this, Greter and colleagues found that “IL-34 did not control the embryonic development of microglia but contributed to microglia homeostasis in the adult” (*Immunity*, 2012). Notably, in Wang’s paper, adult microglia were reduced but not abolished in IL-34 KO mice. This discrepancy in the literature should be considered and discussed.

The reviewer is absolutely correct in making these points. We have adjusted our textual presentation of IL-34 in the introduction and results. Given that the 2 papers (Wang et al, 2012; Greter et al, 2012) show a partial but not complete dependence of microglia on IL-34, our partial dependence is one of the features that highlights notable commonality between PNS and CNS macrophages. However, we agree that, given that other macrophages like Langerhans cells are dependent upon IL-34, this connection is not sufficient for us to say as strongly as we had in earlier versions that PNS macrophages are microglia-like.

4. It is somewhat unexpected and potentially important that macrophages in sciatic nerves of IL-34^{-/-} mice were reduced compared to WT mice, in part likely due to lack of IL-34 expression in Schwann cells (Fig. 5). Is IL-34 also expressed in the DRG? Are DRG macrophages impacted by the loss of IL-34? Does CSF1, another cognate CSF1R ligand, compensate for the loss of IL-34? Notably, CSF1 is detected in Schwann cells of sciatic nerve (Trias et al., *Glia* 2019), constitutively expressed in both satellite cells and sensory neurons in the DRG (Guan et al., *Nat Neurosci* 2016).

The reviewer raises a quite intriguing question here. We are aware of data from nearby collaborators that IL-34 mRNA is expressed in the DRG, in this case by a cell type other than a Schwann cell. However, the work is not ours and is just being prepared for submission and thus we raise this here just for the reviewer’s knowledge.

On the other hand, we are able to add data that quantifies macrophages in the DRG. Indeed, we find that DRG macrophages were also reduced in IL34^{-/-} mice. These new findings are now included as supplemental figure 13. The addition of data allowed us to separately quantify macrophages in the nearby nerve root in the same images. Data from these adjacent locations are shown in the new supplemental figure. We thank the reviewer for the suggestion to look at this issue more closely.

5. What is “subcutaneous fascial nerve” described in this study? There are sensory nerve fibers present in the fascia layer. But I am not aware there is a distinct large nerve which was illustrated in Fig. 1a. Please provide explanation and references.

Our terminology was not the best here. We have updated the terminology used, as the nerve referred to as “fascial” is a cutaneous intercostal nerve. It may be easier to appreciate in this image below, provided for the reviewer’s convenience. The nerves are in red and the left side shows the skin with the right side showing the intercostal muscles. Likely the ones that have recently been described in a recent *Immunity* paper (Kolter et al., *Immunity* 2019), now cited in our manuscript. To clarify this issue, we have edited the manuscript to add the word intercostal to our use of the term “fascial” and we cite the Kolter et al. paper at first inclusion of this nerve type in the manuscript.

6. Flt3-Cre-LSL-YFPfl/fl mice were used for scRNA-seq in Fig. 6. Surprisingly, myelin or neurofilament in sciatic nerve was positive for YFP (Fig. 6g). No additional information was provided in the manuscript to suggest a different mouse line was used in Fig. 6g. Therefore, I will have a serious concern about the specificity of YFP+ myeloid cells sorted for scRNA-seq. A minor point, please amend figure legends which only describes 6 out of 7 panels in the figure.

The reviewer raises an interesting point. In response, we have re-evaluated Flt3Cre-YFP expression versus autofluorescence in nerve sections and confirm that YFP is indeed expressed by neurons (see below, panel a.). Indeed, Flt3 expression has been reported in neurons, where it may play a role in neural stem cell proliferation and survival (Brazel CY, et al. Mol Cell Neurosci. 2001). This is now cited in the manuscript. In light of this evidence, the Flt3-Cre reporter mouse is not only useful for marking hematopoietic stem cells, but seemingly neural progenitors as well.

However, the reviewer's point does raise a potential caveat, which is that PNS macrophages might be falsely labeled as YFP+ if they uptake nerve debris. Here, just to share with the reviewer why we think that is unlikely, we turned to a model in CX3CR1^{GFP} x MPZ-tdTomato mice, where neurons are brightly fluorescent (in the red Tomato channel) and macrophages green fluorescent to see if CX3CR1^{GFP+} PNS macrophages may contain red debris, leaving red to overlap with green fluorescence. We do not see such overlap in homeostasis (below, panel b). By flow cytometry, we only see evidence for strong phagocytosis (red fluorescence in green macrophages) after nerve crush injury (panel c, below). Thus, we argue that nerve material does not strongly contaminate the macrophage pool in nerves.

Furthermore, in our manuscript, we show that CCR2+ macrophages exist only in the YFP+ population. This makes sense given that this population arises from hematopoietic origin. It would be unlikely that only HSC-derived macrophages phagocytose YFP+ nerves. Thus, we are convinced that our Flt3 model and sorting strategy remain valid.

Analysis of the specificity of Flt3-Cre LSL-YFP mice and consideration of possible transfer of fluorescence via uptake of dying cells (related to Figure 6). A. Representative imaging of unstained nerves showing Cre- and Cre+ nerves in Flt3-Cre LSL-YFP mice (Scale bar, 100 μ m). B. Representative imaging of steady state sciatic nerve macrophages in CX3CR1^{GFP/+} MPZ^{tdTomato} (tomato shown in red) mice (Scale bar, 30 μ m). C. Flow cytometric analysis of tdTomato+ myelin uptake in GFP+ sciatic nerve macrophages from contralateral and ipsilateral nerves 3 days following sciatic nerve crush injury.

Minor points:

1. The order of some references and figures listed in the text is often confusing. The following changes are suggested for improvement:

- Page 3/line 48, "during embryogenesis (14) and depend on IL-34 for development (15)";
- Page 4/line 61-62, "CX3CR1-GFP+ cells were located in the endoneurium (Fig 1b) and expressed colony-stimulating factor 1 receptor (CSF1R), also known as CD115 (Fig. 1c).

We have made all these changes in the text.

2. Figs. 1a-c.

a. In Fig. 1b, Col1 staining was used to mark the perineurial layer of sciatic nerve. But Col1 was not introduced in the manuscript.

We have removed depiction of the collagen I staining, as the reviewer is correct that we never referred to it in the manuscript.

b. In the enlargement of the outlined area in Fig. 1c, one CX3CR1-GFP+ cell in the center did not coexpress CSF1R. Again this makes me wonder how specific the sorted cells were.

It is important to keep in mind that these images are cross-sections and that CD115 is primarily localized on the cell surface, with GFP in the cytoplasm. It is possible to have a portion of a macrophage enriched in one or the other. Also, it is important to remember that immunostaining a molecule is usually about one log less sensitive for detection by eye than flow cytometry. These are reasons why the specificity of flow cytometric sorting cannot be construed from fluorescence imaging of tissue cross-sections.

c. To better illustrate macrophages, may change MPZ staining from red to blue.

This is a great suggestion. We have changed CD115 to be shown in red and the MPZ in blue, allowing for clearer examination of green and red overlap.

d. Please describe CX3CR1GFP/+ tdTomato mice in methods.

We have added this strain to the methods.

e. Fig. 1h, please also provide representative images from CD45.2+ tdTomato+ parabiont.

Unfortunately, we cannot provide the requested images, because we used all the CD45.2 parabiont tissue for flow cytometry (see Supplementary figure 1). Indeed, antibodies to distinguish CD45.1 and CD45.2 are challenging to use accurately in tissue sections, but they work very well in flow cytometry.

f. Fig. 1i: % tdTomato+ cells should be plotted individually based on the tissue sources.

In Fig. 1i, different peripheral nerves were combined prior to analysis by flow cytometry to facilitate and ensure sufficient cell quantities for analysis. We have clarified that point in the current version of the manuscript.

3. Fig. 3e, I do not quite understand the rationale to use Clec7a and MHCII staining to rule out the technical artifact in macrophages signature. A MHCIIhi (Lyve1lo) tissue resident macrophage population was already found surrounding the nerves at steady state (Chakarov et al., Science 2019). Therefore, high MHCII expression is neither surprising nor activated microglia-specific.

This staining was to show that these are not induced signatures from sorting and that expression is present in steady state fresh-frozen nerves. We believe that the marker of greater relevance is Clec7a over MHC II. We are aware of the findings of Chakarov and feel that these populations are similar, but cannot rule out the potential difference between macrophages in axonal trunks and associated with nerve endings (in tissue). Since we can show that frozen sections reveal expression of the markers, and frozen tissue has not been subjected to disaggregation, we are certain the expression of these markers is not a disaggregation artifact.

To improve clarity of our statement in the manuscript, we revised the text to read : "Resting PNS macrophages were clearly marked by Clec7a and MHCII (Fig. 3e), suggesting that the signature obtained in PNS macrophages *is not a technical artifact that arose from activation induced by disaggregation of tissue.*"

4. Fig. 5a: Please include FACS plot of CNS microglia as a gating control.

We realize that we failed to refer to Supplemental Figure 10 in the past version of the manuscript. This issue has been corrected. Please refer to Supplementary Figure 10 to find these data.

5. Page 10/line 210: “GFP+ cells were totally absent from Ccr2GFP/GFP mice (Fig. 5e, f)”. However, CCR2+ cells were still detected in 3 out of 5 Ccr2GFP/GFP mice by immunostaining (Fig. 5f). Image quantitation needs to be better explained in Methods.

We changed the text to “almost entirely absent”.

6. IL34 KO mice can be used as a negative control for ISH staining. Please also provide the resource of ISH probes in Methods.

The probes were designed by ThermoFisher with the catalog number - VB1-14592-VT. We have now added this information to the Methods section.

7. In the future, the authors may consider including female mice and expanding their effort in the context of nerve injury.

This suggestion will be certainly be interesting for the future.

REVIEWERS' COMMENTS:

Reviewer #4 (Remarks to the Author):

This resubmission has been significantly improved. The authors have addressed most of my concerns, although I still have a few minor comments.

1. Update the Fig 6 legend in which only 6 (a-f) out of 7 panels (a-g) were described.
2. Delete "Fig 5c" which prematurely appeared in line 195, page 10.
3. Secure a permanent housing for homeless Fig 5d in page 10, 2nd paragraph.
4. Provide the number of sections per DRG and the total number of mice used for quantification of DRG macrophages in Supplementary Fig 13.
5. Thanks for confirming Flt3-YFP expression in sections of sciatic nerve. In agreement with the authors, Flt3-Cre line might be useful to study "neural stem cells proliferation and survival". Therefore, a representative image illustrating YFP expression in the DRG sensory neurons of Flt3-Cre mice, will significantly strengthen the authors' conclusion and facilitate discussion around the potential application of this line. However, I will not insist if the lab is facing some significant challenges during the current rapid evolving COVID outbreak.

Reviewer #4 (Remarks to the Author):

This resubmission has been significantly improved. The authors have addressed most of my concerns, although I still have a few minor comments.

1. Update the Fig 6 legend in which only 6 (a-f) out of 7 panels (a-g) were described.
We have updated the figure 6 legend.

2. Delete “Fig 5c” which prematurely appeared in line 195, page 10.
We have made this revision in the document.

3. Secure a permanent housing for homeless Fig 5d in page 10, 2nd paragraph.
We have made the correction.

4. Provide the number of sections per DRG and the total number of mice used for quantification of DRG macrophages in Supplementary Fig 13.
We have included this in the Supplementary figure 13 legend.

5. Thanks for confirming Flt3-YFP expression in sections of sciatic nerve. In agreement with the authors, Flt3-Cre line might be useful to study “neural stem cells proliferation and survival”. Therefore, a representative image illustrating YFP expression in the DRG sensory neurons of Flt3-Cre mice, will significantly strengthen the authors’ conclusion and facilitate discussion around the potential application of this line. However, I will not insist if the lab is facing some significant challenges during the current rapid evolving COVID outbreak.
We thank the reviewer for this insightful suggestion. Unfortunately, our lab activity is limited at this time and this experiment is not currently feasible.